# Konjac Glucomannan Counteracted the Side Effects of Excessive Exercise on Gut Microbiome, Endurance, and Strength in an Overtraining Mice Model

**DOI:** 10.3390/nu15194206

**Published:** 2023-09-29

**Authors:** Yu-Heng Mao, Minghan Wang, Yu Yuan, Jing-Kun Yan, Yanqun Peng, Guoqin Xu, Xiquan Weng

**Affiliations:** 1School of Exercise and Health, Guangzhou Sport University, Guangzhou 510500, Chinayuanyumail@126.com (Y.Y.); 13610220245@163.com (Y.P.); xugq@gzsport.edu.cn (G.X.); 2Engineering Research Center of Health Food Design & Nutrition Regulation, Dongguan Key Laboratory of Typical Food Precision Design, China National Light Industry Key Laboratory of Healthy Food Development and Nutrition Regulation, School of Life and Health Technology, Dongguan University of Technology, Dongguan 523808, China; jkyan_27@163.com

**Keywords:** polysaccharides, prebiotics, dietary fiber, performance, microbiota

## Abstract

Excessive exercise without adequate rest can lead to overtraining syndrome, which manifests a series of side effects, including fatigue, gut dysbiosis, and decremental sports performance. Konjac glucomannan (KGM) is a plant polysaccharide with numerous health-improving effects, but few studies reported its effects on the gut microbiome, endurance, and strength in an overtraining model. This study assessed the effect of KGM on gut microbiome, endurance, and strength in mice with excessive exercise. Three doses of KGM (1.25, 2.50, and 5.00 mg/mL) were administrated in drinking water to mice during 42 days of a treadmill overtraining program. The results showed that excessive exercise induced a significant microbial shift compared with the control group, while a high dose (5.00 mg/mL) of KGM maintained the microbial composition. The proportion of *Sutterella* in feces was significantly increased in the excessive exercise group, while the moderate dose (2.50 mg/mL) of KGM dramatically increased the relative abundance of *Lactobacillus* and SCFA production in feces. Additionally, the moderate dose and high dose of KGM counteracted the negative effects of excessive exercise on strength or/and endurance (43.14% and 39.94% increase through a moderate dose of KGM, Bonferroni corrected *p* < 0.05, compared with the excessive exercise group). Therefore, it suggests that KGM could prevent overtraining and improve sports performance in animal models.

## 1. Introduction

While the beneficial effects of regular and proper physical activities were extensively studied [1], the side effects of excessive training were underestimated, and the solutions were lacking [2]. Excessive exercise followed by inadequate recovery can lead to overtraining syndrome, which occurs commonly in people who participate in high-volume training, such as competitive athletes, fitness enthusiasts, soldiers, and some other physically active populations [3]. Overtraining manifests a series of physical and psychological side effects, including immunity suppression [4], chronic fatigue [5], gut permeability increase, high risk of infection [6,7], decremental sports performance [8], anxiety, and depression [9]. A few studies also reported that excessive exercise induces gut dysbiosis, including overgrowth of pathogenic bacteria [10].

Excessive exercise can negatively affect both physical and psychological health for physically active populations [7], and some studies reported that several dietary supplements, such as whey protein hydrolysates [11], octacosanol [12], etc., showed an anti-fatigue effect in vivo through different mechanistic pathways. However, few studies reported the effects of prebiotics on gut microbiome and sports performance under the condition of excessive exercise, although the beneficial effects of prebiotics in regulating gut microbiome have been extensively studied in various disease models [13].

Konjac glucomannan (KGM) isolated from the tuber of *Amorphophallus konjac C. Koch.* is a widely used dietary fiber. KGM is a water-soluble polysaccharide with a linear molecular structure that contains (1→4)-β-D-glucopyranose and (1→4)-β-D-mannopyranose with a certain degree (approximately 1 in 19 sugar units) of acetylation at the C-6 position [14]. It is a commercial functional ingredient in the food and nutritional industry for its thickening and gelling properties, as well as its health-improving effects, including weight control, antihyperglycemic, anticancer, and prebiotic effects [14]. KGM was previously reported to show a synergistic effect with exercise on weight loss and body composition [15], but its effect on gut microbiome and sports performance in physically active populations with excessive exercise is unclear.

Our previous studies showed that native KGM was more effective in protecting the gut bacteria against antibiotic perturbance and modulating the gut microbiome compared with the degraded KGM fractions with lower molecular weights [16,17,18]. Therefore, we hypothesized that the supplementation of KGM would modulate the microbial composition, facilitate physiological adaption, and, meanwhile, enhance sports performance in mice with excessive training. Therefore, this study aims to investigate the effects of KGM on gut microbiome and sports performance in mice with excessive training. Additionally, the dose effect of KGM will also be estimated in this study. This study could indicate the potential application of KGM in preventing overtraining for athletes, soldiers, and other physically active populations.

## 2. Materials and Methods

### 2.1. Chemicals

The konjac glucomannan (KGM) used in this study was purchased from Johnson Co. (Ezhou, China) and was characterized in our previous in vivo study [17]. The molecular weight of KGM used in the study is 1.82 × 10^7^ Da with an intrinsic viscosity of 9.48 dL/g, and it is composed of mannose and glucose at the ratio of 1.65:1. Eight short-chain fatty acid standards, including acetic acid, propionic acid, *n*-butyric acid, *iso*-butyric acid, *n*-valeric acid, *iso*-valeric acid, *n*-hexanoic acid, and *iso*-hexanoic acid, were all purchased from Aradin (Shanghai, China). The DNA extraction kit was obtained from Tiangen Biotech (Beijing, China). Appendix A lists all compounds used in this study with detailed information from the NCBI PubChem compound database and the supplier sources.

### 2.2. Animal Experiment Design

Thirty male (8-week-old) C57BL/6J mice were purchased from SPF (Beijing) Biotechnology Co., Ltd. (Beijing, China) and were randomly divided into five groups (*n* = 6 per group, one cage for each group) and had free access to a standard chow (AIN-93G purified diet, the formula is provided in Appendix A) and water during the experiment. Only male mice were used in the study to avoid the interference of hormones on sports performance. As shown in Figure 1a, mice in the control group (Ctl) and the excessive exercise group (EX) were fed with sterilized drinking water for 42 days. Referring to our previous study [17], mice in the low-dose KGM group (KGM-L), moderate-dose KGM group (KGM-M), and high-dose KGM group (KGM-H) groups were fed with a sterilized KGM solution at 1.25, 2.50, and 5.00 g/L as drinking water, respectively, for 42 days. The enriched water and the food were refreshed every two or three days. Mice of the EX, KGM-L, KGM-M, and KGM-H groups underwent 42 days of overtrained exercise program (Table 1). Mice were trained by a ZH-PT/5S treadmill (Anhui Zhenghua biological apparatus facilities, Huaibei, China), and the overtrained exercise program was set according to a reported study with slight modification [19]. The intensity, incremental degree, and training time were all increased gradually in the 42 days (6 weeks). The mice were trained for 5 continuous days followed by 2 days of recovery during each experimental week.

Mice were raised in a specific pathogen-free (SPF) facility (Guangzhou Sport University Animal Center) under standard conditions (22–24 °C, 50 ± 5% humidity) for a 12 h light/dark cycle. All animals were raised in accordance with the National Research Council’s Guide (NRC 2011) for the Care and Use of Laboratory Animals. The animal experiment was approved by the Animal Experimental Ethics Inspection of Guangzhou Sport University (Permit No. 2022DWLL-24).

The food and water intake were recorded every week (on days 7, 14, 21, 28, 35, and 42), and body weight was recorded every 2 weeks (on days 0, 14, 28, and 42). The feces of each mouse were collected with sterilized tubes weekly for gene sequencing and short-chain fatty acid (SCFA) determination. On day 42, the maximum grip force and endurance test were conducted for all mice to assess their sports performance. Then, the mice were anesthetized, and the blood was collected by heart puncture in a tube containing acid citrate dextrose solution. The whole blood was analyzed immediately by a BC-5130 auto hematology analyzer (Shenzhen Mairui, Shenzhen, China) for red blood cell (RBC) counting and hemoglobin (HGB) determination, followed by centrifugation at 4000 rpm (Sorvall ST 8R small benchtop centrifuge, Thermo Fisher Scientific, Osterode am Harz, Germany) for 15 min to obtain plasma for creatine kinase (CK) and free hemoglobin (FHb) and short-chain fatty acid (SCFA) determination. All mice were sacrificed after blood collection, and the small intestine and colon were dissected for weight, length measurements, and intestinal morphology observation.

### 2.3. Maximum Grip Force Test

The maximum grip force test was conducted by an XR501 Grip Strength Meter (Shanghai XINRUAN Information Technology Co., Ltd., Shanghai, China). Mice were stretched parallelly on a force-meter rod using their forelimb, and the tail was pulled in the opposite direction until the grasp was broken [20]. Mice were tested 5 consecutive times, and the maximum value was recorded.

### 2.4. Endurance Test

The aerobic endurance performance was assessed by increasing the treadmill speed from an initial speed (10 m/min) with a fixed 10° inclination in 1 m increments every 1 min until exhaustion [21]. The point of exhaustion was defined as the time at which the mouse reached the electric grid ≥ 5 times in 1 min. The running time until exhaustion was recorded.

### 2.5. Intestinal Morphology

The colon was washed with saline, fixed in 4% paraformaldehyde solution for 24 h at 4 °C, embedded in paraffin, and then stained with hematoxylin and eosin (H&E) staining. The histological examination was conducted using the Pannoramic 250 FLASH system (3DHISTECH Ltd., Budapest, Hungary), and the pathological changes were analyzed by CaseViewer Software 2.4.

### 2.6. Determination of Fatigue-Associated Biomarkers in Plasma

RBC counting and HGB were determined by a BC-5130 auto hematology analyzer (Shenzhen Mairui, Shenzhen, China) as mentioned above. CK and FHb were quantified by using commercial kits (Nanjing Jiancheng Bioengineering Institute, Nanjing, China) according to the instructions from the manufactories. Triplicates were conducted for each sample.

### 2.7. Determination of Short-Chain Fatty Acids in Plasma and Feces

The short-chain fatty acids (SCFAs) in plasma and feces were determined by gas chromatography-mass spectrometry (GC-MS) according to a reported study with slight modification [22]. Eight SCFA standards used for identification and quantification included acetic acid, propionic acid, *n*-butyric acid, *iso*-butyric acid, *n*-valeric acid, *iso*-valeric acid, *n*-hexanoic acid, and *iso*-hexanoic acid (Aladdin^®^, Shanghai, China). *N*- and *iso*-butyric acid, *n*- and *iso*-valeric acid, and *n*- and *iso*-hexanoic acid were combined to be butyric acid, valeric acid, and hexanoic acid in the final calculation, respectively. In brief, plasma and feces were completely homogenized with 0.05% phosphoric acid (1:1, *v*/*v* or *w*/*v*) by a vortex mixer; then, HPLC grade ethyl acetate (100 μL for plasma and 400 μL for feces) was added followed by vortex mixing for 3 min and centrifuged at 10,000 rpm by a 5424 R centrifuge (Eppendorf, Hamburg, Germany) for 10 min at 4 °C. The supernatant was collected and filtered by a 0.22 μm membrane for injection. An Agilent 7890B GC system combined with an Agilent 7000D triple quadrupole mass spectrometry (Agilent Technologies Inc., Santa Clara, CA, USA) was applied for the determination of SCFAs. A capillary column (dimension 30 × 0.25 mm, Agilent DB-WAX, Agilent) was applied for gas phase separation. The carrier gas was helium at a flow rate of 1 mL/min. The initial column temperature (40 °C) was increased to 95 °C at the rate of 40 °C/min and kept for 1 min, then increased to 140 °C at the rate of 5 °C/min and kept for 1 min, then increased to 200 °C at the rate of 40 °C/min and kept for 1 min, and finally increased to 240 °C at the rate of 20 °C/min and kept for 4 min. Both the injection temperature and the interface temperature were 250 °C. The electron bombardment ion source temperature was 230 °C. The electron ionization source was set at 70 eV, and ions were detected by the full scan monitoring method. The injection volume was 1 μL.

### 2.8. DNA Extraction and 16S rRNA Sequencing

The genomic DNA extraction was conducted by Tiangen stool DNA extraction kit (Tiangen, Beijing, China) and followed the manufacturer’s manual. The DNA concentration of the microorganism was quantified by NanoDrop 2000 (Thermo Fisher, Waltham, MA, USA). The 16S rRNA gene sequencing was performed by Wekemo Technology Co., Ltd. (Shenzhen, China). All data were demultiplexed and adapter trimmed using the DADA2 plugin in Qiime2 software and formed an Amplicon Sequence Variant (ASV). Then, the species annotation was obtained by being referenced to Greengenes Database 13_8.

### 2.9. Statistical Analysis

The repeated measures analysis of variance (ANOVA) was applied for body weight at the end of the experiment. One-way ANOVA with the least-significant difference (LSD) or Tamhane T2 post hoc test was applied to analyze the difference in body weight among groups at each time point [23], intestinal morphological indicators, the maximum grip force, running time, fatigue-associated biomarkers, concentrations of SCFAs in plasma, diversity, and taxonomic relative abundances among different groups. As the primary endpoint was sports performance, a Bonferroni correction for multiple testing [24] was applied as five groups were involved in statistical analysis for the parameters of sports performance. Two-way analysis of variance (MANOVA) was used to analyze the concentrations of SCFAs in feces [25]. Permutational multivariate analysis of variance (PERMANOVA) was applied to analyze the Bray–Curtis distance. For the microbial function prediction data with multi-step transformation processing, one-way ANOVA with the Duncan post hoc test was applied according to a previous report [26]. For those data that did not fulfill normal distribution, the Kruskal–Wallis test was applied. *p* < 0.05 was considered to be a significant difference. The statistical analyses were performed by SPSS 16.0. The Bray–Curtis dissimilarity principal component analysis (PCA) was used to assess microbial variation among fecal samples by QIIME 2.

## 3. Results

### 3.1. The Influence of KGM on Body Weight, Water and Food Intake

Figure 1b–d shows the effect of KGM on body weight (Figure 1b), water intake (Figure 1c), and food intake (Figure 1d) in mice. In general, the body weight of mice increased gradually in 42 days in different groups, despite some fluctuation, and there was no significant difference among the different groups at the end of the experiment (Figure 1b). As shown in Figure 1c, excessive exercise with or without KGM supplementation increased the water intake at most time points compared with the Ctl. In contrast, excessive exercise decreased food intake at days 14 and 21 (Figure 1d), which might be due to the appetite suppression induced by the accumulative exercise load [27], while increased compensatorily at days 28, 35, and 42 (Figure 1c,d) after gradual adaptation [28]. Supplementation of KGM with different doses relieved the appetite suppression induced by excessive exercise.

### 3.2. The Influence of KGM on Sports Performance

Figure 2 shows the influence of excessive exercise and KGM supplementation on sports performance in mice. Compared with the Ctl, excessive exercise significantly decreased the maximum grip force (Figure 2a) and running time (Figure 2b). Different doses of KGM supplementation significantly increased the maximum grip force and running time compared with the EX group (Figure 2a,b), while no dose-effect response was found among different doses for both strength and endurance (Figure 2a). The KGM groups showed no significant difference in both strength and endurance compared with the Ctl group; however, KGM increased the tolerance to excessive exercise and enhanced sports performance compared with the EX group.

### 3.3. The Influence of KGM on Blood Biochemical Indices

Figure 3 shows the effect of excessive exercise and KGM on biochemical indices in blood. HGB indicates the training load, and CK indicates the intensity of exercise. These two indices were commonly used to assess the exercise load and intensity for monitoring the training load for athletes [29]. The red blood cell (RBC) counts (Figure 3a) and hemoglobin (HGB) (Figure 3b) in the EX group significantly decreased compared with the Ctl group. Similarly, excessive exercise also induced a significant increase in the creatine kinase (CK) concentration in plasma compared with the Ctl group. Both RBC and HGB were significantly increased by different doses of KGM compared with the EX group, but no dose-effect response was found (Figure 3a,b). The supplementation of KGM significantly decreased the CK concentration with a general dose-effect response (Figure 3c) compared with the EX group.

Moreover, Figure 3d shows that the free hemoglobin (FHb) in the EX group was significantly increased compared with the Ctl group, which indicated that excessive exercise damaged the RBC and resulted in high free hemoglobin in the plasma. Compared with the EX group, KGM showed a significant effect on protecting RBC (Figure 3d).

### 3.4. Intestinal Morphography

Figure 4 presents the effect of excessive exercise and KGM on intestinal length and weight. The weight of the total intestine was significantly (ANOVA, *p* < 0.05) increased by excessive exercise compared with the Ctl group, while KGM at a low dose further increased the weight of the total intestine compared with the EX group (Figure 4b). The lengths of the total intestine (Figure 4a) and the small intestine (Figure 4c) were significantly decreased in the KGM-L group, and KGM at low and moderate doses increased the weight of the small intestine significantly by approximately 20% (Figure 4d) compared to the EX group, which might be related to the need of nutrient absorption [30]. Additionally, there was no significant difference found in the length and weight of the colon among the different groups (Figure 4e,f).

The H&E staining pictures in Figure 4g show the effect of excessive exercise and KGM on intestinal morphology. The epithelial structure of the Ctl group was well structured without obvious inflammatory infiltration. Excessive exercise induced prominent inflammatory infiltration with a great number of lymphocytes infiltrating and gathering between the mucosal muscle and basal layer. According to the histological scoring system (Appendix A), the histopathologic score in Figure 4h shows that all different doses of KGM showed significant anti-inflammatory effects and maintained the micro-structure of colonic epithelium with a dose-effect response.

### 3.5. The Influence of KGM on Gut Microbiome

Generally, excessive exercise and KGM supplementation induced a significant shift in the gut microbiome. Figure 5 shows the alpha-diversity indices of gut microbiome in all groups on day 42. The different alpha-diversity indices reflect microbial richness and evenness. The observed_otus is an index referring to the number of operational taxonomic units (OTUs) measured in the sample. The Faith’s phylogenetic diversity (faith’s_pd) is calculated based on the total branch lengths of a phylogenetic tree. The larger the faith’s_pd, the greater the evolutionary difference, in other terms, the greater the diversity. The chao1 is commonly used to estimate the total number of species. It is a nonparametric indicator reflecting a theoretical richness from the observed indices. The Shannon’s index indicates the species richness of the community. The Simpson’s index is used to estimate the similarity of the community, and it reflects the dominance based on accounting proportion of species in the community. The rarefaction curves of the alpha diversity on days 0, 21, and 42 are shown in Appendix A, respectively.

Compared with the Ctl, 21 days of excessive exercise showed no significant effect on the alpha-diversity indices (Appendix A), because the incremental intensity exercise program was applied. The supplementation of a low dose of KGM decreased the chao1 (Appendix A) and observed_otus (Appendix A), while moderate and high doses of KGM showed no effect on the alpha diversity. After 42 days of excessive exercise, the chao1, faith_pd, observed features, and Shannon index of gut microbiome were significantly decreased compared with the Ctl group (Figure 5a–d).

The relative abundances of the top 20 taxa in feces at different levels on day 21 are shown in Appendix A. Excessive exercise induced limited alteration of the gut microbiome. At the phylum level, the three most abundant taxa (Bacteroidetes, Firmicutes, and Proteobacteria) remained unchanged. Only the relative abundance of Tenericutes was increased by three times compared with the Ctl, and the supplementation of KGM counteracted the changes induced by excessive exercise. Compared with the EX group, the supplementation of different doses of KGM increased the relative abundance of Bacteroidetes significantly (58.29% with EX versus 66.00% with KGM-L versus 66.46% with KGM-M versus 68.58% with KGM-H). At the genus level, excessive exercise significantly increased the relative abundance of *Prevotella* (4.58% with Ctl versus 10.2 with EX) and *AF12* (0.17 with Ctl versus 1.46 with OT), and the supplementation of three doses of KGM counteracted this alteration. Compared with the EX group, a low dose of KGM significantly increased the proportion of *Bacteroides* (8.71% with Ctl versus 14.41% with KGM-L), and both moderate and high doses of KGM significantly increased the relative abundances of *Prevotellaceae_Prevotella* (2.69% with EX versus 6.72% with KGM-M versus 8.21% with KGM-H).

On day 42, the gut microbiome in the EX group and KGM groups showed an obvious shift compared with the Ctl (Figure 6), and the Bray–Curtis distance from the EX and KGM groups to the Ctl group on day 42 was enlarged compared with day 0 (Appendix A) and day 21 (Appendix A). At the phylum level (Figure 6a and Table 2), the relative abundance of *Bacteroidetes* dramatically increased from 36.92% in the Ctl group to 61.34% in the EX group. The supplementation of KGM maintained this change (48.06% for KGM-L, 60.14% for KGM-M, and 62.07% for KGM-H) with no significant differences among the different doses. In contrast, the relative abundance of *Firmicutes* significantly decreased from 49.91% (Ctl) to 29.57% (IE) after 42 days of excessive exercise, and all the KGM groups maintained the relative abundance of *Firmicutes* from 28.94% to 37.08% (compared with EX, *p* > 0.05, ANOVA).

At the family level (Figure 6b and Table 2), the relative abundance of S24_7 was significantly increased from 26.21% in the Ctl group to 40.58% in the EX group. The low-dose supplementation of KGM decreased the relative abundance of S24_7, while the moderate and high doses of KGM maintained their relative abundance compared with the EX group. Additionally, excessive exercise significantly (*p* < 0.05, ANOVA) increased the relative abundances of Bacteroidaceae (2.22% with Ctl versus 5.97% with EX), Alcaligenaceae (0.68% with Ctl versus 3.01% with EX), and Prevotellaceae (1.01% with Ctl versus 3.51% with EX) at the expense of unclassified taxa (12.02% with Ctl versus 8.13% with EX), Lachnospiraceae (11.21% with Ctl versus 6.74% with EX), and Ruminococcaceae (7.14% with Ctl versus 3.02% with EX). In addition, there were no significant differences in the proportion of most taxa among three doses of KGM at the family level. Exceptionally, only a moderate dose of KGM (13.79%) dramatically increased the relative abundance of Lactobacillaceae compared to the Ctl (1.17%), EX (2.03%), KGM-L (3.97%), and KGM-H (2.69%) groups.

At the genus level (Figure 6c and Table 2), excessive exercise significantly increased the proportion of *Sutterella* from 0.67% (Ctl) to 3.01% (IE) and decreased the proportion of *Ruminococcaceae_Ruminococcus* and other taxa by approximately 50%. The low, moderate, and high doses of KGM differently affected the relative abundances of major taxa compared with the EX group. The moderate dose of KGM dramatically increased the relative abundance of *Lactobacillus* by nearly seven times compared with the EX group, while no significant difference was found among the Ctl, EX, KGM-L, and KGM-H groups. On one side, the low dose of KGM significantly increased the relative abundance of *Allobaculum* by more than three times (7.16% with Ctl versus 22.08% with KGM-L). On the other side, in an obvious trade-off in numerical dominance within the gut microbiome, the relative abundances of *Prevotella* (8.46% with EX versus 3.78% with KGM-L) and *Coprococcus* (1.11% with EX versus 0.33% with KGM-L) were significantly decreased by KGM-L. Additionally, both KGM-M and KGM-H significantly decreased the proportion of *Bifidobacterium* compared with the EX group (1.26% with EX versus 0.07% with KGM-M versus 0.22% with KGM-H). The relative abundance of the top 20 taxa in feces at different levels on day 42 is shown in Appendix A.

As shown in Figure 6d,e, compared with the Ctl group, excessive exercise induced a significant shift of microbial composition (Bray–Curtis distance = 0.51, permanova-pairwise test, *p* < 0.05) on the horizontal direction (Axis 2, 26.89%). The supplementation of high-dose KGM slightly but significantly shortened the shift (Bray–Curtis distance from Ctl to KGM-H was 0.49, Bray–Curtis distance = 0.51, permanova-pairwise test, *p* < 0.05) induced by excessive exercise oi this direction (Axis 2). Notably, the two lower doses of KGM induced an obvious shift of microbial composition in the vertical direction (Axis 2, 19.93%), which resulted in a larger Bray–Curtis distance to the Ctl (0.58 with KGM-L and 0.57 with KGM-M).

Additionally, the time effect of each treatment was also evaluated. Appendix A shows the Bray–Curtis distances from days 0, 21, and 42 to day 0 in five groups. At the end of the study, all groups showed a significant microbial shift from the starting point, and the Bray–Curtis distance between day 0 and day 42 was larger than the Bray–Curtis distance between day 0 and day 21 in all five groups. Notably, the supplementation of KGM decreased the Bray–Curtis distance between day 0 and day 21 (0.67 with EX, 0.62 with KGM-L, 0.61 with KGM-M, and 0.53 with KGM-H), as well as the Bray–Curtis distance between day 0 and day 42 (0.71 with EX, 0.71 with KGM-L, 0.69 with KGM-M and 0.59 with KGM-H) with a dose-effect relationship. Additionally, the Bray–Curtis distance between days 0 and 21 was generally larger than the alteration between day 21 and day 42 in the exercise and KGM groups.

Additionally, Figure 7 shows the metabolic changes in the gut microbiome induced by excessive exercise and KGM supplementation on day 42. The microbial genes were annotated to the Kyoto Encyclopedia of Genes and Genomes (KEGG) orthology (KOs). The 42 days of excessive exercise and KGM supplementation caused obvious changes to microbial function at level 2 and level 3.

On the KEGG level 2, excessive exercise induced 15 KEGGs changes, among which 9 KEGGs were decreased, including “carbohydrate metabolism”, “amino acid metabolism”, “energy metabolism”, etc., and 6 KEGGs were increased, including “cell growth and death”, “glycan biosynthesis and metabolism”, “transport and catabolism”, etc. Generally, the supplementation of KGM further decreased or increased the function, especially at the moderate dose. The top 20 pathways in the KEGG level 2 are shown in Figure 7a, among which the KOs with significant differences among different groups are shown in Figure 7c. Among the top 20 KOs, “carbohydrate metabolism”, “amino acid metabolism”, “global and overview maps”, “energy metabolism”, “membrane transport”, “cell motility”, and “cellular community–prokaryotes” were decreased significantly after 42 days of excessive exercise, while the “glycan biosynthesis and metabolism” and “cell growth and death” were increased compared with the Ctl group. Compared with the EX group, a moderate dose of KGM significantly influenced several KEGGs, including further decreasing “amino acid metabolism”, “global and overview maps”, and “energy metabolism”, while increasing “metabolism of other amino acids” and “lipid metabolism”. Comparatively, the supplementation of KGM at low or high doses showed no significant difference compared with OT.

On the KEGG level 3, excessive exercise induced 80 KOs changes in total, among which 41 KOs were significantly decreased and 39 KOs were increased compared with the Ctl on day 42. Figure 7b shows the top 20 pathways in the KEGG level 3, among which the KOs with significant differences among different groups are shown in Figure 7d. Excessive exercise significantly decreased “valine, leucine, and isoleucine biosynthesis”, “biosynthesis of amino acids”, “carbon fixation in photosynthetic organisms”, “pantothenate and CoA biosynthesis”, and “lysine biosynthesis”. Compared with the EX group, moderate-dose KGM further decreased “valine, leucine, and isoleucine biosynthesis”, “biosynthesis of amino acids”, “carbon fixation in photosynthetic organisms”, and “pantothenate and CoA biosynthesis”. In contrast, “other glycan degradation” was significantly increased in all groups with excessive exercise (IE, KGM-L, KGM-M, and KGM-H). Additionally, the moderate-dose KGM increased “other glycan degradation” the most.

### 3.6. Short-Chain Fatty Acids

Figure 8 shows the generation of short-chain fatty acids in feces (Figure 8a–f) and plasma (Figure 8g) during 42 days in five groups. Acetic acid, propionic acid, and butyric acid were the major SCFAs in feces, while acetic acid and hexanoic acid were the major SCFAs in plasma. Specifically, the individual and total SCFAs in feces were maintained at similar levels in the Ctl group at different time points (Figure 8a–f), and no significant difference was found among the different groups on day 0. The individual and total SCFAs accumulated gradually in the EX and KGM groups during 42 days, although no significance was found in propionic acid, butyric acid, valeric acid, hexanoic acid, and total SCFAs because of the large variation in individuals in different-dose KGM groups on day 14 and 28. Notably, the low-dose KGM induced a significant (*p* < 0.05, ANOVA) increase in acetic acid on day 14, but this difference disappeared (*p* > 0.05, ANOVA) on day 28. This might be due to the different adaptation times of the gut microbiome to different intake levels of dietary fiber, which is consistent with the gut microbial shift from day 0 to 42 (Figure 6e). On day 42, the moderate dose of KGM significantly enhanced the acetic acid and total SCFAs by approximately 100% compared with the other four groups (Figure 8a–f). For plasma (Figure 8g), no significant difference was found in the five individual SCFAs and total SCFAs among the different groups in plasma because of the large variation among the individuals.

## 4. Discussion

In this study, the mode of intervention and dose of KGM in drinking water was considered comprehensively. Because data on the dietary intake of athletes were limited, we took the guidelines from the American Heart Association [31] into consideration instead. The recommended daily intake (RDI) of dietary fiber for an adult is 25–30 g; however, there is approximately 10–15 g of deficiency from the actual average intake to the RDI [32,33]. According to Meeh’s formula [34], 1500–2250 mg KGM per kg body weight per day for mice was suggested when KGM was assumed to be the only extra fiber source. Even though the maximum gavage volume (300 μL, 3 times per day) was used, it was impossible to achieve the dose target as the maximum concentration of KGM solution (with high viscosity) because the gavage needle was 2 mg/mL. Therefore, 5 g/L was used as the maximum concentration in the drinking water to ensure the accessibility for the mice. According to the water intake (Figure 1c), the actual intake of KGM was 6.56, 14.03, and 22.90 mg/day/mice in the KGM-L, KGM-M and KGM-H groups, respectively. Approximately 250–900 mg per kg body weight per day was achieved. Even though it was still lower than the dose target (1500–2250 mg fiber per kg body weight per day), it was consistent with the range (0–800 mg per kg body weight per day) reported in mice with dietary fiber intervention [35], and it was also acceptable for considering other possible dietary fiber sources.

The results in Figure 1b–d reflected the bifacial effects of KGM on food intake and body weight. As a soluble dietary fiber, KGM suppresses the appetite by enhancing satiety under normal conditions [36], while low and moderate doses of KGM alleviate the appetite suppression induced by excessive exercise (Figure 1c,d). This indicates that there might be a delicate balance in appetite promotion and suppression by KGM depending on the dose of supplementation and the energy homeostasis of the subjects.

HGB is an important index to assess exercise load, while CK reflects the exercise intensity. The significant decrease in HGB and the significant increase in CK reflected the high exercise load and intensity of the training program and the successful construction of the overtraining model [37]. In addition, HGB was closely related to exercise ability, especially aerobic capacity (endurance) [38]. The significant increase in FHb in the EX group (Figure 3d) was consistent with the decrease in RBC counting (Figure 3a), and this could partially explain the significant decrease in running time (endurance) in the EX group (Figure 2b). KGM significantly increased the tolerance to excessive exercise and enhanced sports performance compared with the EX group (Figure 2b); it even failed to improve the sports performance better than the Ctl group (only equal to the Ctl group), which was due to the substantial damage caused by excessive exercise. Moreover, it referred to the effect of KGM on improving physical function and sports performance under a well-designed training plan and adequate recovery. The effect of KGM on improving sports performance also proposed its possible influence on chronic pathologies that are characterized by a reduction in the quality of life and the development of fatigue [39].

Some studies reported that the intake of dietary fiber usually increased microbial diversity because more taxa were involved in digesting the complicated carbohydrates through cross-feeding [40]. However, compared with the EX group, the supplementation of KGM generally decreased the alpha diversity further with the dose-effect response. One of the explanations might be that the increasing energy requirement induced by exercise exaggerated the competitive inhibition of the major taxa participating in carbohydrate metabolism (energy recollection) against other minor taxa. Additionally, KGM at a high dose showed an inhibitory effect on the growth of some taxa as previously reported [41].

The effects of excessive exercise and KGM on gut microbiome also altered over time. On day 21, limited changes were found in the EX group, while the supplementation of KGM significantly increased the relative abundance of Bacteroidetes at the phylum level and *Prevotellaceae_Prevotella* at the genus level compared with the EX group. Bacteroidetes was the major phylum utilizing undigestible and complicated carbohydrates [42]. *Prevotellaceae_Prevotella* contains the necessary enzyme and gene clusters for the fermentation and utilization of complicated carbohydrates; therefore, the relative abundance of *Prevotellaceae_Prevotella* was increased to recollect more energy from the intestinal content to meet the increasing energy requirement induced by exercise.

On day 42, this trend became more obvious, and excessive exercise significantly increased the proportion of Bacteroidaceae, Alcaligenaceae, and Prevotellaceae, which were all reported to be key taxa for degrading complicated carbohydrates (glycans), and their relative abundances were boosted because of the increasing energy requirement induced by excessive exercise, which is consistent with previously reported studies [43]. In addition, only a moderate dose of KGM dramatically increased the relative abundance of Lactobacillaceae. This phenomenon might be related to both the physical and the biological properties of KGM. KGM could be fermented by gut bacteria as prebiotics; however, a high concentration of KGM with high viscosity showed a significant effect in inhibiting the growth of bacteria [16]. Only a moderate dose of KGM could be utilized by gut microbiota while showing limited effect on the growth rate of gut bacteria, thereby showing the best beneficial effect on gut microbiome under disturbed conditions such as overtraining. This is similar to our previous study [17]. At the genus level, excessive exercise significantly increased the relative abundance of *Sutterella*, which is the most abundant genus in *Proteobacteria* and a core commensal genus in the gastrointestinal tract and was reported to show slight pro-inflammatory effects [44]. The supplementation of KGM increased the relative abundance of *Allobaculum*, which is reported to produce butyric acid and negatively correlated with inflammation, insulin resistance, and obesity in mice [45]. This could also partially explain the effect of KGM in reducing inflammation in colonic tissue (Figure 4g,h). On the other side, the relative abundances of *Prevotella* and *Coprococcus* were decreased by *KGM-L. Prevotella* is a large genus containing many various strains. *Prevotella* strains are linked with chronic inflammation, although they are associated with plant-rich dietary patterns in the meanwhile [46]. *Coprococcus* plays an important role in maintaining or improving microbial homeostasis in the host through interacting synergistically with the beneficial endogenous bacteria and presenting antipathogenic effects, acting by competitive inhibition, improvement of epithelial barrier function, and secretion of antimicrobial substances [47]. Additionally, KGM with higher concentrations decreased the relative abundance of *Bifidobacterium*, which was consistent with our previous report in vitro that a high dose of KGM exhibited a growth-inhibitory effect on bifidobacteria [16]. Although the supplementation of high-dose KGM significantly shortened the microbial shift induced by excessive exercise (Figure 6e), we are not able to conclude that KGM-H was the best solution for the present study because it is not like the previous study on the gut microbiota with perturbance of antibiotics [17] in which we aimed to minimize the microbial shift from the Ctl (baseline); the effect of exercise with different intensity was not black and white.

On the one hand, excessive exercise induced a significant change in gut microbial composition; however, no significant difference was found in the individual and total SCFA concentration in feces and plasma compared with the Ctl group. This might be due to the circumstance that the SCFA generation induced by excessive exercise might be metabolized quickly. On one hand, SCFAs contribute to energy homeostasis via multiple cellular metabolic pathways [48]. On the other hand, SCFAs could be utilized directly as additional energy substrates for exercise [49]. The microbial function prediction also showed that “other glycan degradation” was increased by excessive exercise, while “amino acid metabolism”, “energy metabolism”, and “carbohydrate metabolism” were significantly decreased by excessive exercise. Therefore, by combining the gut microbial shift, microbial function prediction, and SCFA generation, it was suggested that the gut microbiome played an important role in energy re-harvesting to satisfy the dramatic increase in energy induced by excessive exercise at the cost of some other metabolic function.

Both the moderate- and high-dose KGM enhanced the sports performance (Figure 2), while their effect on the gut microbial composition differed (Figure 6). This might be due to the circumstance that the major mechanisms of KGM-M and KGM-H on enhancing sports performance were different. KGM-M increased “the other glycan degradation” and increased the generation of SCFAs significantly, which contributed to the energy re-harvesting and thus sports performance enhancement. In addition, acetic acid and butyric acid were reported to activate the biogenesis of mitochondria via the AMPK-PGC-1α pathway [50,51], which might consequently affect energy metabolism and sports performance. KGM-H showed no significant effect on increasing SCFAs compared with the EX group. This might be due to the circumstance that the water and food intake increased in the last 14 days in the KGM-H group (Figure 1c,d) before the sports performance test was taken. Importantly, combining the previous studies [16,17], KGM-H was able to maintain the stability of the gut microbiome and increase the biofilm formation of gut bacteria, finally improving intestinal integrity.

All these findings indicated the comprehensive effect of KGM on gut microbiome and sports performance. On one side, the lower dose of KGM induced the most significant microbial shift (Figure 6d,e) and increased SCFA generation (Figure 8), contributing to energy re-collection. On the other side, the higher dose of KGM maintained microbial stability at the cost of fast bacterial growth. Additionally, the influence of KGM on appetite and nutrient absorption should also be considered. Therefore, to make a trade-off between stimulating beneficial bacterial growth and maintaining microbial stability, the moderate dose showed the best effects on gut microbial composition and function, as well as sports performance under excessive exercise.

Additionally, the great variance in microbial composition and SCFA generation within KGM groups also indicated that the response of gut microbiota to complicated dietary fibers, such as KGM, varies greatly among different individuals, which should be considered in future studies. In addition, KGM at a low dose increased acetic acid at day 14, but the difference disappeared on days 28 and 42, which indicated the microbial adaptation to KGM. Another study also reported a similar phenomenon, i.e., that continuous supplementation of inulin did not further increase the SCFA generation [52].

Finally, the species difference should be taken into consideration, and the whole genomics should be used in future studies to investigate the key taxa modulating the gut microbiome and metabolisms, especially under complicated conditions. Moreover, germ-free mice could be applied to illustrate the causal relationship between gut microbiome and sports performance.

## 5. Conclusions

In conclusion, excessive exercise induced a notable gut microbial shift and functional alteration and decreased sports performance in mice. The supplementation of moderate- and high-dose KGM significantly enhanced strength and endurance, and the moderate-dose KGM showed the best effect on counteracting the side effects of excessive exercise while maintaining the benefits of exercise, including improving the microbial composition and the SCFA generation. These results suggested that KGM could be a useful strategy for enhancing the tolerance of excessive exercise and preventing overtraining, as well as improving sports performance. The underlying mechanism is probably related to energy re-collection by gut microbiota, and the key taxa involved in this process need to be further deeply studied.

## Figures and Tables

**Figure 1 nutrients-15-04206-f001:**
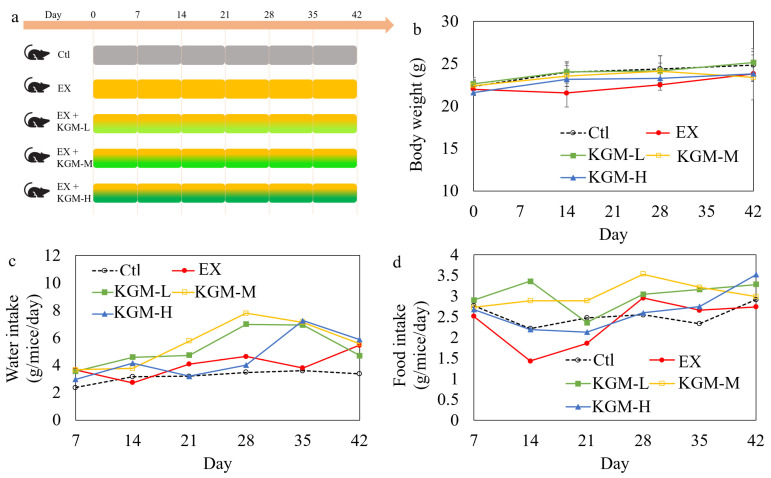
The flow chart of the animal experiment (**a**), the body weight (**b**), water intake (**c**), and food intake (**d**) of the different groups during the 42 days. Average of six samples; error bars represent the standard deviation at *n* = 6. No error bars were generated for (**c**,**d**) because the average water and food intake was calculated weekly in total. Not having the same letters indicates the significant difference among different groups for the same index; repeated measures ANOVA was applied for the body weight at the end of the experiment, and one-way ANOVA with LSD or Tamhane T2 post hoc test was performed for comparing body weight among groups at various time points, *p* < 0.05. Ctl: control; EX: excessive exercise; KGM-L: low dose of konjac glucomannan (KGM) (1.25 g/L in drinking water) with excessive exercise; KGM-M: moderate dose of KGM (2.50 g/L in drinking water) with excessive exercise; KGM-H: high dose of KGM (5.00 g/L in drinking water) with excessive exercise.

**Figure 2 nutrients-15-04206-f002:**
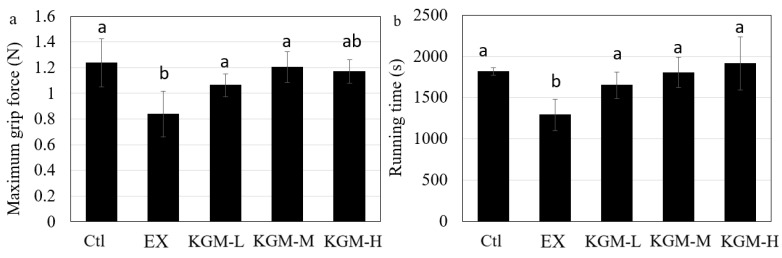
The effect of different doses of konjac glucomannan (KGM) on sports performance in mice on day 42. (**a**) Maximum grip force; (**b**) endurance time. Average of six samples; error bars represent the standard deviation at *n* = 6. Not having the same letters indicates the significant difference among different groups for the same index, one-way ANOVA with Bonferroni post hoc test, *p* < 0.05. Ctl: control; EX: excessive exercise; KGM-L: low dose of KGM (1.25 g/L in drinking water) with excessive exercise; KGM-M: moderate dose of KGM (2.50 g/L in drinking water) with excessive exercise; KGM-H: high dose of KGM (5.00 g/L in drinking water) with excessive exercise.

**Figure 3 nutrients-15-04206-f003:**
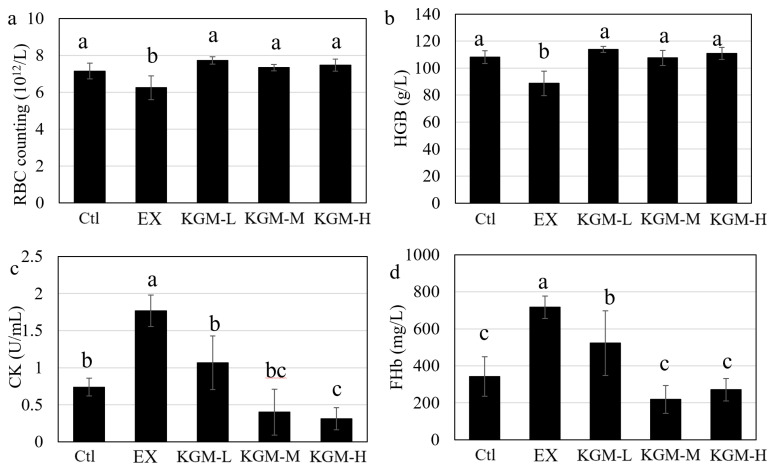
The influence of different doses of konjac glucomannan (KGM) on biochemical indices in blood on day 42. (**a**) Red blood cell (RBC) counts and (**b**) hemoglobin (HGB) in whole blood; (**c**) creatine kinase (CK) and (**d**) free hemoglobin (FHb) in plasma. Average of six samples; error bars represent the standard deviation at *n* = 6. Not having the same letters indicates the significant difference among different groups for the same index, one-way ANOVA with LSD or Tamhane T2 post hoc test, *p* < 0.05. Ctl: control; EX: excessive exercise; KGM-L: low dose of KGM (1.25 g/L in drinking water) with excessive exercise; KGM-M: moderate dose of KGM (2.50 g/L in drinking water) with excessive exercise; KGM-H: high dose of KGM (5.00 g/L in drinking water) with excessive exercise.

**Figure 4 nutrients-15-04206-f004:**
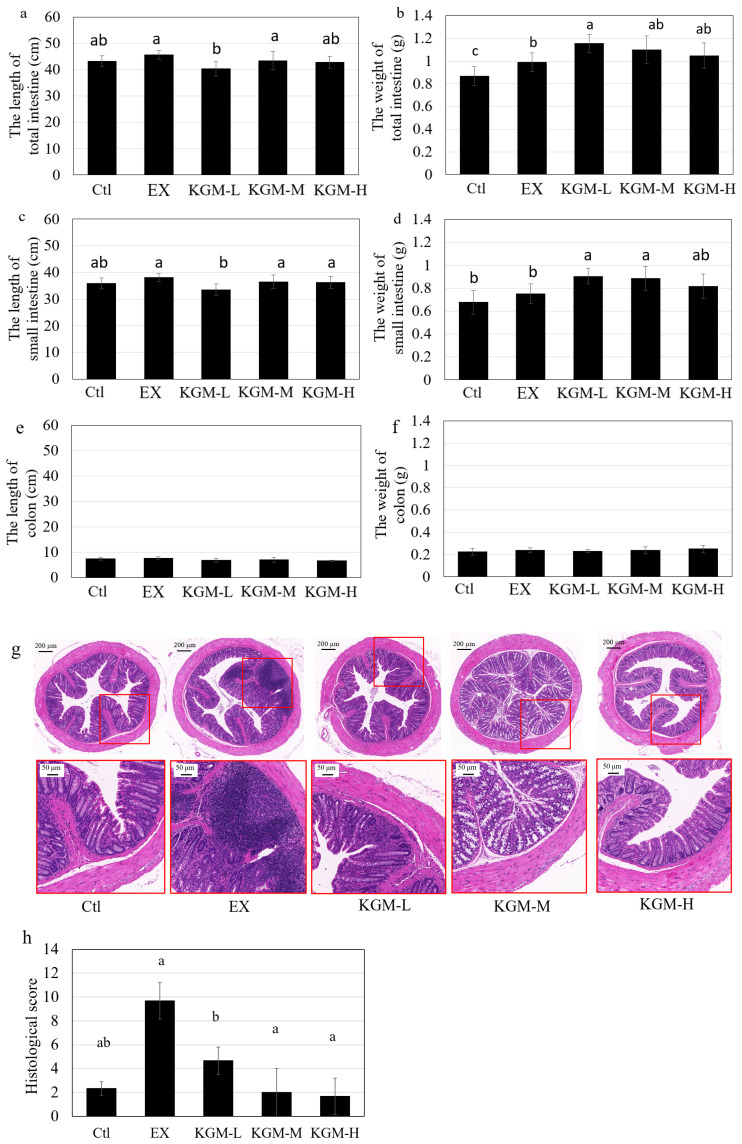
The influence of different doses of konjac glucomannan (KGM) on the colonic morphography in mice. The length of the (**a**) total intestine, (**c**) small intestine, and (**e**) colon. The weight of the (**b**) total intestine, (**d**) small intestine, and (**f**) colon. The representative pictures of the colon with H&E staining (**g**), and the histological score (**h**) for different groups. Average of six samples; error bars represent the standard deviation at *n* = 6. Not having the same letters indicates the significant difference among different groups for the same index, one-way ANOVA with LSD or Tamhane T2 post hoc test, *p* < 0.05. Ctl: control; EX: excessive exercise; KGM-L: low dose of KGM (1.25 g/L in drinking water) with excessive exercise; KGM-M: moderate dose of KGM (2.50 g/L in drinking water) with excessive exercise; KGM-H: high dose of KGM (5.00 g/L in drinking water) with excessive exercise.

**Figure 5 nutrients-15-04206-f005:**
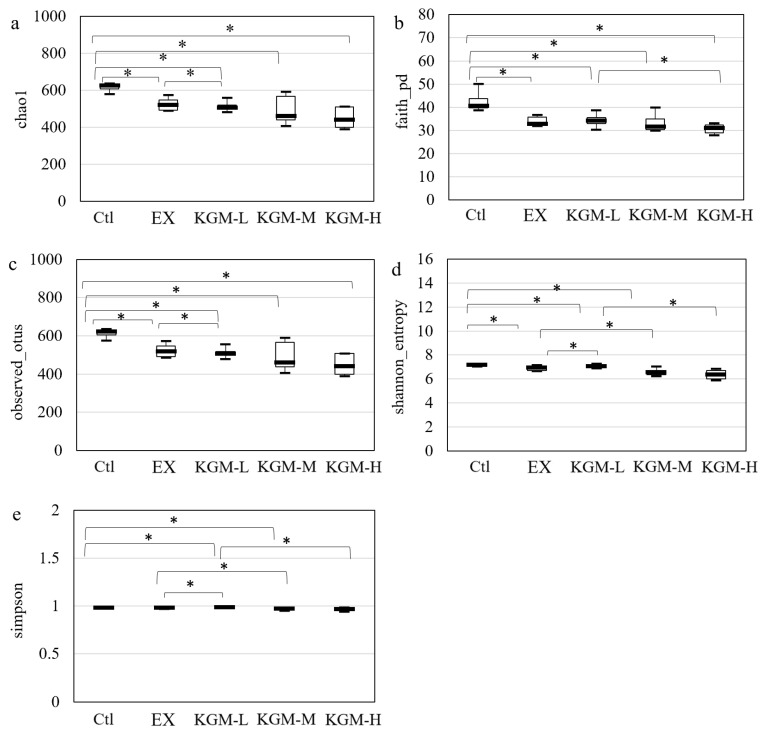
The alpha-diversity indices of fecal microbiome on day 42. (**a**) chao1, (**b**) Faith’s phylogenetic diversity (faith_pd), (**c**) observed_otus, (**d**) shannon_entropy index, (**e**) Simpson index. The star indicates the significant difference between the two groups. Asterisks indicate the significant difference between the two groups, Krustal–Wallis test, *p* < 0.05, *n* = 6. Ctl: control; EX: excessive exercise; KGM-L: low dose of KGM (1.25 g/L in drinking water) with excessive exercise; KGM-M: moderate dose of KGM (2.50 g/L in drinking water) with excessive exercise; KGM-H: high dose of KGM (5.00 g/L in drinking water) with excessive exercise.

**Figure 6 nutrients-15-04206-f006:**
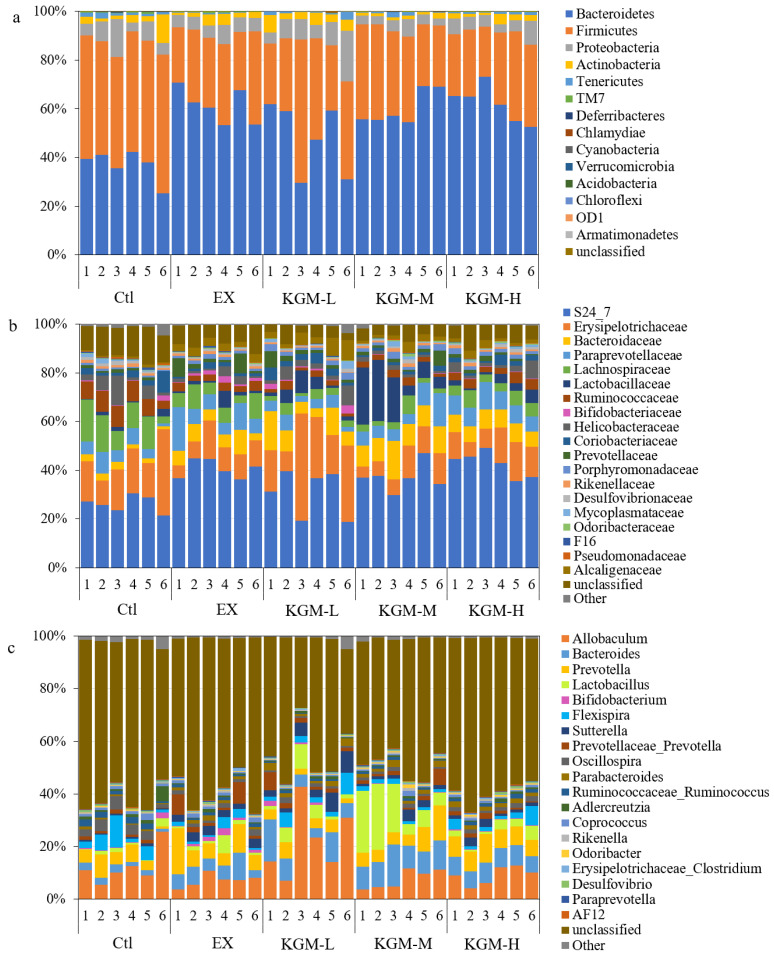
The effects of different doses of (KGM) on the fecal microbial composition on day 42 at the (**a**) phylum level, (**b**) family level, and (**c**) genus level. (**d**) The Bray–Curtis distances between the Ctl group and the five groups; (**e**) the principal component analysis (PCoA) on day 42. Not having the same letters indicates a significant difference between the two groups, PERMANOVA, *p* < 0.05, *n* = 6. Ctl: control; EX: excessive exercise; KGM-L: low dose of KGM (1.25 g/L in drinking water) with excessive exercise; KGM-M: moderate dose of KGM (2.50 g/L in drinking water) with excessive exercise; KGM-H: high dose of KGM (5.00 g/L in drinking water) with excessive exercise.

**Figure 7 nutrients-15-04206-f007:**
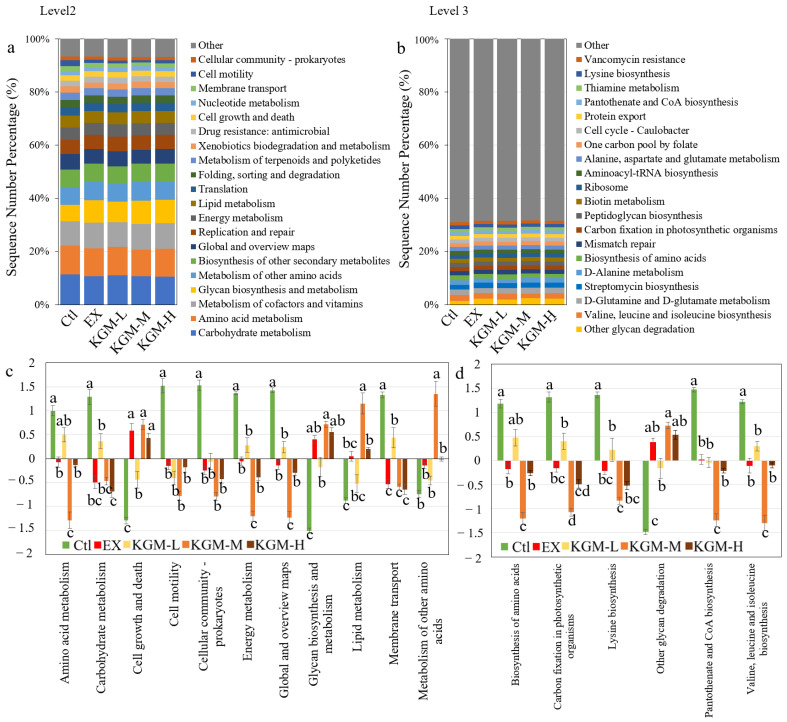
The effects of different doses of KGM on the fecal microbial functional PICRUSt function analysis of mice on day 42. The bar chart of the top 20 predicted functions (**a**) at level 2 and (**b**) at level 3. The bar chart of the abundance of the functions with significant differences among different groups (**c**) at level 2 and (**d**) at level 3. Average of six samples; error bars represent the standard deviation at *n* = 6. Not having the same letters indicates a significant difference by one-way ANOVA with the Duncan post hoc test, *p* < 0.05. Ctl: control; EX: excessive exercise; KGM-L: low dose of KGM (1.25 g/L in drinking water) with excessive exercise; KGM-M: moderate dose of KGM (2.50 g/L in drinking water) with excessive exercise; KGM-H: high dose of KGM (5.00 g/L in drinking water) with excessive exercise.

**Figure 8 nutrients-15-04206-f008:**
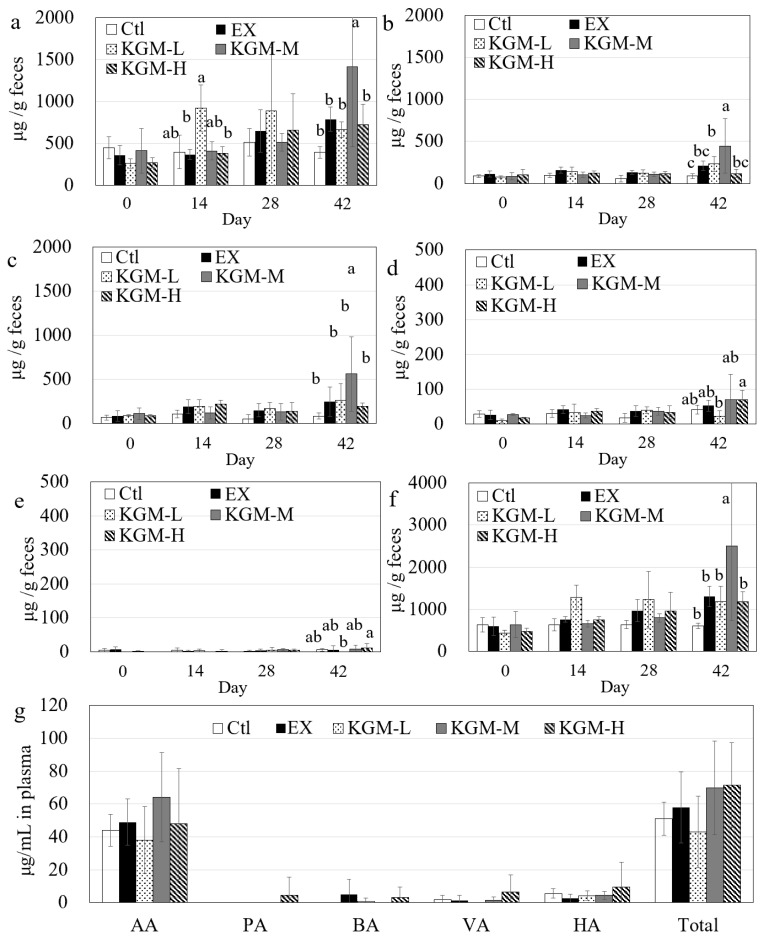
The effect of excessive exercise and KGM on SCFAs in feces and plasma in mice. The effects of different doses of KGM on the concentration of (**a**) acetic acid (AA), (**b**) propionic acid (PA), (**c**) butyric acid (BA), (**d**) valeric acid (VA), (**e**) hexanoic acid (HA), and (**f**) total SCFAs in feces at days 0, 14, 28, and 42. Not having the same letters indicates a significant difference between groups at the same time point by the two-way analysis of variance with Tukey post hoc tests, *p* < 0.05; (**g**) individual and total SCFAs in plasma at day 42. Average of six samples; error bars represent the standard deviation at *n* = 6. Ctl: control; EX: excessive exercise; KGM-L: low dose of KGM (1.25 g/L in drinking water) with excessive exercise; KGM-M: moderate dose of KGM (2.50 g/L in drinking water) with excessive exercise; KGM-H: high dose of KGM (5.00 g/L in drinking water) with excessive exercise.

**Table 1 nutrients-15-04206-t001:** The protocol of the overtrained exercise program.

Week	Intensity (% EV)	Duration (min)	Sessions/Day	Recovery between Sessions (h)	Treadmill Grade (up, %)
1	60	30	1	24	0
2	60	45	1	24	0
3	60	60	1	24	14
4	70	60	1	24	14
5	75	75	1	24	14
6	75	75	2	4	14

Note: EV: exhaustion velocity, the exhaustion velocity was determined by increasing the treadmill speed from an initial speed (6 m/min) with 0% inclination in 3 m increments every 3 min until exhaustion, EV = V + (n/b) × a. The time point of exhaustion was defined as the time at which the mouse reached the electric grid ≥ 5 times in 1 min; V: the speed of the last completed session; n: lasting time in the incomplete session; b: lasting time per session (3 min); a: speed increment of the treadmill (3 m/min).

**Table 2 nutrients-15-04206-t002:** Major taxa with significant alteration at different levels on day 42. Ctl: control; EX: excessive exercise; KGM-L: low dose of KGM (1.25 g/L in drinking water) with excessive exercise; KGM-M: moderate dose of KGM (2.50 g/L in drinking water) with excessive exercise; KGM-H: high dose of KGM (5.00 g/L in drinking water) with excessive exercise.

	Ctl	EX	KGM-L	KGM-M	KGM-H
**Phylum**					
Bacteroidetes	36.92 ± 6.16 b	61.34 ± 7.15 a	48.06 ± 14.59 ab	60.14 ± 7.03 a	62.07 ± 7.54 a
Firmicutes	49.91 ± 3.89 a	29.57 ± 5.82 b	37.08 ± 12.69 b	33.16 ± 6.39 b	28.94 ± 5.88 b
Proteobacteria	7.47 ± 4.42 ab	5.71 ± 1.10 ab	9.33 ± 5.87 a	4.29 ± 1.59 b	5.74 ± 2.27 ab
Actinobacteria	3.79 ± 3.91 ab	2.64 ± 1.43 ab	4.26 ± 1.78 a	1.34 ± 0.53 b	2.25 ± 1.24 ab
Cyanobacteria	0.01 ± 0.01 ab	0.00 ± 0.00 b	0.01 ± 0.00 a	0.02 ± 0.03 ab	0.02 ± 0.01 ab
unclassified	0.02 ± 0.01 a	0.01 ± 0.01 bc	0.00 ± 0.00 c	0.01 ± 0.01 ab	0.01 ± 0.01 bc
**Family**					
S24_7	26.21 ± 3.36 c	40.58 ± 3.76 a	30.64 ± 9.47 bc	37.06 ± 5.62 ab	42.48 ± 5.22 a
unclassified	12.02 ± 1.98 a	8.13 ± 2.29 b	4.44 ± 1.97 b	4.80 ± 1.09 b	6.83 ± 1.88 b
Bacteroidaceae	2.22 ± 0.77 b	5.97 ± 2.50 a	8.24 ± 4.71 ab	10.41 ± 2.88 a	7.13 ± 0.66 a
Paraprevotellaceae	5.12 ± 2.65 ab	8.69 ± 5.05 a	3.98 ± 1.65 b	7.08 ± 3.68 ab	7.96 ± 1.69 ab
Lactobacillaceae	1.17 ± 1.09 b	2.03 ± 2.36 b	3.97 ± 3.27 b	13.79 ± 9.69 a	2.69 ± 1.80 b
Ruminococcaceae	7.14 ± 2.02 a	3.02 ± 0.83 b	2.09 ± 0.96 b	2.37 ± 0.84 b	3.40 ± 1.03 b
Alcaligenaceae	0.68 ± 0.23 b	3.01 ± 0.62 a	4.55 ± 2.75 ab	2.04 ± 1.33 ab	1.87 ± 1.07 ab
Prevotellaceae	1.01 ± 0.49 b	3.51 ± 3.46 a	2.44 ± 2.14 ab	2.06 ± 2.10 ab	1.24 ± 0.64 ab
Porphyromonadaceae	0.67 ± 0.21 b	1.29 ± 0.37 ab	1.84 ± 0.84 ab	1.76 ± 0.73 ab	1.67 ± 0.38 a
Rikenellaceae	1.04 ± 0.20 a	0.88 ± 0.22 ab	0.54 ± 0.11 b	1.02 ± 0.40 ab	0.94 ± 0.27 ab
Desulfovibrionaceae	1.08 ± 0.41 a	0.61 ± 0.28 ab	0.50 ± 0.39 b	0.60 ± 0.50 ab	0.77 ± 0.39 ab
Bifidobacteriaceae	0.53 ± 0.76 ab	1.26 ± 0.89 ab	1.22 ± 1.25 ac	0.07 ± 0.07 b	0.22 ± 0.07 b
**Genus**					
unclassified	59.56 ± 6.53 ab	58.68 ± 7.26 ab	43.57 ± 11.57 ab	47.87 ± 5.45 b	58.80 ± 4.25 a
*Bacteroides*	2.22 ± 0.77 b	5.97 ± 2.50 ab	8.24 ± 4.71 ab	10.41 ± 2.88 a	7.13 ± 0.66 a
*Lactobacillus*	1.17 ± 1.09 b	2.03 ± 2.36 b	3.97 ± 3.27 b	13.79 ± 9.68 a	2.69 ± 1.80 b
*Sutterella*	0.67 ± 0.23 b	3.01 ± 0.62 a	4.55 ± 2.75 ab	2.04 ± 1.33 ab	1.87 ± 1.07 ab
*Parabacteroides*	0.67 ± 0.21 b	1.29 ± 0.37 ab	1.84 ± 0.84 ab	1.76 ± 0.73 ab	1.67 ± 0.38 a
*Ruminococcaceae* *_Ruminococcus*	1.95 ± 0.77 a	0.98 ± 0.24 bc	0.60 ± 0.35 c	0.73 ± 0.26 bc	1.17 ± 0.42 b
*Coprococcus*	1.23 ± 0.59 a	1.11 ± 0.44 a	0.33 ± 0.15 b	0.69 ± 0.69 ab	0.70 ± 0.20 ab
*Bifidobacterium*	0.53 ± 0.76 ab	1.26 ± 0.89 a	0.66 ± 0.70 ab	0.07 ± 0.07 b	0.22 ± 0.07 b
*AF12*	0.28 ± 0.10 ab	0.23 ± 0.07 a	0.10 ± 0.03 b	0.19 ± 0.03 a	0.21 ± 0.04 a
Other	2.13 ± 1.40 a	0.75 ± 0.21 b	1.36 ± 1.78 ab	1.06 ± 0.62 ab	0.70 ± 0.19 ab

Data are shown in average ± standard deviation. Not having the same letters indicates the significant difference among different groups for the same index, one-way ANOVA with LSD or Tamhane T2 post hoc test, *p* < 0.05, *n* = 6.

## Data Availability

Data available on request from the authors.

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
