# Peer review of "Konjac Glucomannan Counteracted the Side Effects of Excessive Exercise on Gut Microbiome, Endurance, and Strength in an Overtraining Mice Model"

_nutrients, 2023, doi:10.3390/nu15194206_

Round 1
Reviewer 1 Report
General comments:
The paper investigates the effect of excessive exercise and Konjac glucomannan in mice on sport performance and gut microbiome in mice. They find that excessive exercise leads to worse sport performance and changes in gut mirobioma and to increase in fatigue-associated markers and to inflammation of the gut. The application of Konjac glucomannan counteracts the negative effects of excessive exercise on sport performance and gut inflammation, but the effects on the gut microbiome are not consistent.
The definition of EX = Exhaustion velocity should be explained better. The formular is not clear to the reader.
Statistics: What was the primary endpoint of the study?
Was there a correction for testing multiple parameters?
Discussion: The finding that KGM has a dose -dependant effect on grip strength and running time but not on gut microbiome. The authors provide several explanations, which are mostly an assumption.
This casts some doubt on the causal relationship between sport performance and gut microbioma.
Specific comments:
Line 85: The sentence must be grammatically corrected, for example by inserting “and” after “ China)
Line 115: The Abbreviation SCFA should be explained
Line 147: The abbreviations CK (creatinine kinase?) and FHb (?) should be explained.
Figure 1: the error bars are missing in fig. 1c and 1d. Significance of difference is not shown in the Figure.
Figure 2 and 3 and 4 and table 2: It is not explained, which letters (a, b, c, ab) designate which significance level.
Line 375 and 376 : The text refers to Fig A.5 and A.6 , which are not shown.
Line 476: Fig. A.7a-e: Does this refer to Fig 7?
If these figures are shown only in the supplementary Appendix, this should be told in the text.
Author Response
General comments:
The paper investigates the effect of excessive exercise and Konjac glucomannan in mice on sport performance and gut microbiome in mice. They find that excessive exercise leads to worse sport performance and changes in gut mirobioma and to increase in fatigue-associated markers and to inflammation of the gut. The application of Konjac glucomannan counteracts the negative effects of excessive exercise on sport performance and gut inflammation, but the effects on the gut microbiome are not consistent.
Response: Thank you for your opinion. Combining the present results and our previous studies, KGM with lower or higher concentrations (with different viscosity) might affect the gut microbiome from different ways, and it was discussed comprehensively in Discussion.
The definition of EX = Exhaustion velocity should be explained better. The formular is not clear to the reader.
Response: The definition was revised to “EV = V + (n/b) × a. … V: The speed of the last completed session”. (Line 112-114)
Statistics: What was the primary endpoint of the study?
Was there a correction for testing multiple parameters?
Response: Thank you for the valuable suggestion on the statistical analysis.
- The primary endpoint was sport performance, and it was defined according to your suggestion. Revised to “The primary endpoint was sport performance.” in line 192-193.
- The statistical analysis was reperformed according to your precious suggestion. Revised to “a Bonferroni correction for multiple testing [24] was applied as five groups were involved in statistical analysis” in line 193-194.
Discussion: The finding that KGM has a dose -dependant effect on grip strength and running time but not on gut microbiome. The authors provide several explanations, which are mostly an assumption.
This casts some doubt on the causal relationship between sport performance and gut microbioma.
Response: Thank you so much for your valuable opinion, and we agree with you. Therefore, further studies using germ-free mice combined with whole genomics will be conducted in the future to explore the deeper causal relationship between gut microbiome and sport performance.
Revised to “Moreover, germ-free mice could be applied for illustrating the causal relationship between gut microbiome and sport performance.” in line 625-627.
Specific comments:
Line 85: The sentence must be grammatically corrected, for example by inserting “and” after “ China)
Response: Revised accordingly in line 86.
Line 115: The Abbreviation SCFA should be explained
Response: Revised to “short chain fatty acid (SCFA)” in line 118-119.
Line 147: The abbreviations CK (creatinine kinase?) and FHb (?) should be explained.
Response: The abbreviations CK and FHb had been explained above in line 126.
Figure 1: the error bars are missing in fig. 1c and 1d. Significance of difference is not shown in the Figure.
Response: There are no error bars for Fig. 1c and 1d, because the average water intake and food intake per mice per day was calculated weekly, not daily, in total (The total food or water intake per week / the number of mice per group). Therefore, the data only showed the general trend of food and water intake during 42 days.
Figure 2 and 3 and 4 and table 2: It is not explained, which letters (a, b, c, ab) designate which significance level.
Response: In order to involve more statistical information, we adopted this way to label the significance among various groups according to our previous report (https://doi.org/10.1016/j.carbpol.2021.118546.) and some other reports (10.3390/nu14163308ï¼›10.3390/nu12041145). For example, “ab” means there is no statistical difference compared with either “a” or “b”. The explanation had been stated in the legend, but we still improved it to be easier to be understood according to your suggestion.
Revised to “Without same letters indicate…” in the legends of Fig. 2, 3, 4, 6, 7 and 8 and Table 2.
Line 375 and 376: The text refers to Fig A.5 and A.6 , which are not shown.
Response: The phrase of “in the supplementary Appendix” was added for all figures shown only in the supplementary Appendix.
Line 476: Fig. A.7a-e: Does this refer to Fig 7?
If these figures are shown only in the supplementary Appendix, this should be told in the text.
Response: Thank you for the suggestion. Revised accordingly.

Reviewer 2 Report
Dear Authors
As one of the reviewers, I express my personal scientific opinion on your work. I would like to reassure you that I was trying to be positive and constructive but particularly as fair and honest as possible to your work. First of all, well done for the whole project and for the time spent to accomplish this task. The good and logical flow in the Introduction and the explanation provided in Method’s section is appreciated. I should also note that the originality of the study and the work done on images, tables and figures are all positive points. However, the lack of the calculation of the Effect Size and CI and test-retest reliability, especially for the performance tests, are somewhat negative points.
Please accept my judgment with a positive and constructive way.
General comments:
1. In your abstract you concluded that “KGM could be used as a supplement to prevent overtraining and improve sport performance”. How sure you are that this particular supplement has exactly the same effect in humans? In my point of view, it should be better to add that KGM may prevent overtraining and improve sports performance “in animal model.”
2. In your Introduction also, you reported that “This study could indicate the potential application of KGM in preventing overtraining for athletes, soldiers, and other physically active population.” Why you did not examine (also) humans?
3. In my point of view, the Results and Discussion section produces study exhaustion. You have provided too many results and figures particularly and therefore too many information in the discussion. By doing this, you have unintentionally reduced perhaps the quality of the comprehensive meaning of the study. Is there any possibility to reduce the content of the paper and present some of the independed variables in an alternative paper? I am just wondering.
4. The article is in general well-written and well-presented. However, could you please check spelling, grammar and syntax in some cases throughout the manuscript for clarity?
Abstract:
5. If it is possible, some important results/numbers and p values should be reported into the abstract.
6. In several cases, the preposition “of” is missing prior to KGM.
Introduction:
7. Lines 45-48: Please check the English grammar for clarity; activate or active, mechanism or mechanistic?
8. Line 61: “was” or is?
9. Line 67: Please consider to place the preposition “of” prior to KGM.
Methods:
10. Were the animals, during the 42 days of the study period, isolated individually or in groups?
11. In the Statistical Analysis, you reported that you used the LSD and Tamhane's T2 as post-hoc tests. Why you are using two post-hoc tests?
12. Since you have 7 timepoints (day 0 – day 42), why you did not perform ANOVA repeated measures to analyze body weight, water and food intake.
13. Which variables did you analyze with Kruskal-Wallis test and why?
Results:
14. Figure 1: In the figure caption you are talking about error bars. Where are the error bars?
15. In the Statistical Analysis, you reported that you used the LSD and/or the Tamhane's T2 as post-hoc tests. However, in the results section, Figure 1 caption, you are mentioning Tukey or Tamhane’s t2 tests; and in Figure 7 you reported Ducan post-hoc test. So LSD and Tamhane's T2 or Tukey and Tamhane's T2, or Ducan? Please clarify. In addition, please explain why you are using various post-hoc statistical tests.
Discussion:
16. You claimed that KGM (L, M, or H) supplementation counteracts the negative consequences that the Excessive Exercise may progressively produce in strength and endurance. However, looking carefully your Figure 2a and b, there is a clear indication that the control group performed better in maximum grip force than any other group and also better in running time than EX and KGML groups and equal with KGMM and KGMH. May we suggest that instead of someone doing regular excessive exercise is better not to do exercise at all, since both strength and endurance will not be deteriorated?
The article is in general well-written and well-presented. However, some minor to moderate editing of English language required.
Reviewer 3 Report
I consider this article is very interesting and well constructed. However I suggest small changes to improve it.... The English is good but there are small inaccuracies that require revision. Furthermore, I think it can be confusing to read the results and the discussion together. I propose to separate the two sections (using subparagraphs) and correct the bibliography. Furthermore, I suggest addressing in the discussion the possible influence of Konjac glucomannan in chronic pathologies which are characterized by a reduction in the quality of life and the development of fatigue.
Minor editing of English language required
Author Response
Reviewer 3: Comments and Suggestions for Authors
I consider this article is very interesting and well constructed. However I suggest small changes to improve it.... The English is good but there are small inaccuracies that require revision. Furthermore, I think it can be confusing to read the results and the discussion together. I propose to separate the two sections (using subparagraphs) and correct the bibliography. Furthermore, I suggest addressing in the discussion the possible influence of Konjac glucomannan in chronic pathologies which are characterized by a reduction in the quality of life and the development of fatigue.
Response: Thank you so much for your generally positive judgment on our study. We made the following modification according to your precious comments to further improve the quality of the manuscript.
- We checked the manuscript carefully throughout again, and all the changes were highlighted in red.
- The Results and Discussion were separated according to your advice, and the bibliography was reordered as well.
- Revised to “The effect of KGM on improving sport performance also proposed its possible influence in chronic pathologies which are characterized by a reduction in the quality of life and the development of fatigue [35]” in line 520-522.

Round 2
Reviewer 2 Report
2nd Revision
I am not very convinced with your response to my point 16 comment. Simply, why the control group performed better in maximum grip force than any other group and also better in running time than EX and KGML groups and equal with KGMM and KGMH? Based on your response to my comment, you are implying that the KGM groups were perhaps all over-trained due to the training-treatment effect; and this is the reason why the control group performed better in the above fitness tests. Am I right? So, if yes:
a) based on your results which variable, that you had collected, may imply overtraining to your KGM groups?
b) Please discuss clearly a bit in your discussion section why the control group performed better in the above fitness test.
Author Response
I am not very convinced with your response to my point 16 comment. Simply, why the control group performed better in maximum grip force than any other group and also better in running time than EX and KGML groups and equal with KGMM and KGMH? Based on your response to my comment, you are implying that the KGM groups were perhaps all over-trained due to the training-treatment effect; and this is the reason why the control group performed better in the above fitness tests. Am I right? So, if yes:
- a) based on your results which variable, that you had collected, may imply overtraining to your KGM groups?
- b) Please discuss clearly a bit in your discussion section why the control group performed better in the above fitness test.
Response: Yes, you are right. In the present study, the Ctl group did perform equally to (not better after a Bonferroni correction according to your advice in the first round revision) three KGM groups.
- The significant decrease of hemoglobin and increase of creatine kinase in Figure 3b-c showed that the excessive training program decreased the physical function.
- Thank you for the suggestion. Revised to “HGB is an important index to assess exercise load, while CK reflects the exercise intensity. The significant decrease of HGB and significant increase of CK reflected the high exercise load and intensity of the training program, and the successful construction of overtraining model [37]. Besides, HGB was closely related to exercise ability, especially aerobic capacity (endurance) [38]. The significant increase of FHb in the EX group (Fig. 3d) was consistent with the decrease of RBC counting (Fig. 3a), and this could partially explain the significant decrease of running time (endurance) in the EX group (Fig. 2b). KGM significantly increased the tolerance to excessive exercise and enhanced sport performance compared with the EX group (Fig. 2-b), even failed to improve the sport performance better than the Ctl group (only equal to the Ctl group), which due to the substantial damage caused by excessive exercise. Moreover, it referred the effect of KGM on improving physical function and sport performance under a well-designed training plan and adequate recovery. The effect of KGM on improving sport performance also proposed its possible influence in chronic pathologies which are characterized by a reduction in the quality of life and the development of fatigue [39].” in line 517-531.